# Qualification of the Microsatellite Instability Analysis (MSA) for Bladder Cancer Detection: The Technical Challenges of Concordance Analysis

**DOI:** 10.3390/ijms25010209

**Published:** 2023-12-22

**Authors:** Thomas Reynolds, Katie Bertsche, David Moon, Chulso Moon

**Affiliations:** 1NEXT Bio-Research Services, LLC, 11601 Ironbridge Road, Suite 101, Chester, VI 23831, USA; 2HJM Cancer Research Foundation Corporation, 10606 Candlewick Road, Lutherville, MD 21093, USA; 3BCD Innovations USA, 10606 Candlewick Road, Lutherville, MD 21093, USA; 4Department of Otolaryngology-Head and Neck Surgery, The Johns Hopkins Medical Institution, Cancer Research Building II, 5M3, 1550 Orleans Street, Baltimore, MD 21205, USA

**Keywords:** bladder cancer, loss of heterozygosity, microsatellite instability, triplet MSA assay

## Abstract

Bladder cancer (here we refer to transitional carcinoma of bladder) is a major cause of morbidity and mortality in the Western world, and recent understanding of its etiology, the molecular characteristics associated with its progression, renders bladder cancer an ideal candidate for screening. Cystoscopy is invasive and sometimes carries unwanted complications, but it is the gold standard for the detection of bladder cancer. Urine cytology, while the most commonly used test as an initial screening tool, is of limited value due to its low sensitivity, particularly for low-grade tumors. Several new “molecular assays” for the diagnosis of urothelial cancer have been developed over the last two decades. Here, we have established our new bladder cancer test based on an assay established for the Early Detection Research Network (EDRN) study. As a part of the study, a quality control CLIA/College of American Pathology (CAP) accredited laboratory, (QA Lab), University of Maryland Baltimore Biomarker Reference Laboratory (UMB-BRL), performed quality assurance analysis. Quality assurance measures included a concordance study between the testing laboratory (AIBioTech), also CLIA/CAP accredited, and the QA lab to ensure that the assay was performed and the results were analyzed in a consistent manner. Therefore, following the technical transfer and training of the microsatellite analysis assay to the UMB-BRL and prior to the initiation of analysis of the clinical samples by the testing lab, a series of qualification studies were performed. This report details the steps taken to ensure qualification of the assay and illustrates the technical challenges facing biomarker validation of this kind.

## 1. Introduction

Microsatellite instability (MSI) is a molecular genotype resulting from genomic hypermutability and is initially discovered as genome-wide variations in the length of microsatellite sequences. As part of familiar colon cancer syndrome, MSI is commonly observed among individuals with Lynch syndrome [1,2,3]. Recently, since its initial discovery, MSI has been recognized and generalized as a phenomenon among a wide spectrum of sporadic cancers [4,5,6,7,8], and the underlying mechanisms for these cases of sporadic cancers seem to be unrelated with inherited genetic mutations, though they are based on epigenetic mechanisms, namely methylation of MLH1. For both inherited and sporadic sold tumors, the most deleterious outcome of MSI would be the accumulation of frameshift mutations in tumor-associated genes, which then can pave crucial pathways leading to human carcinogenesis—the development of cancer. Molecular diagnosis of MSI is relatively simple and currently can be accomplished by examining PCR products amplified by a few (typically 5–7) informative microsatellite markers based on polymerase chain reaction (MSI–PCR) [9,10].

Bladder cancer, the majority of which is transitional carcinoma of the bladder, is one of the major causes of morbidity and mortalities among elderly patients in the Western world [11]. While our understanding of its etiology, molecular characteristics associated with disease progression, and management guidelines have progressed, early detection is still tricky and renders bladder cancer one of the most critical and ideal candidates for early disease screening [12,13,14,15]. The urine cytology test, one of the most commonly used screening methods, is of limited value due to its low sensitivity, particularly to low-grade tumors, though it does provide high specificity. Cystoscopy has been the gold standard for detecting bladder cancer for decades. However, it is an invasive test and often carries unwanted complications, and the relatively high cost can sometimes limit its use. New molecular tests designed to detect both qualitative and quantitative changes among cellular and subcellular alterations exclusively associated with bladder cancer have been explored [14,15,16], and several new “molecular assays” for diagnosing urothelial cancer have been developed over the last two decades. Overall, several molecular assays have been used in conjunction with traditional screening methods and have provided some promising results [16,17,18,19,20]. Based on two different histopathology and clinical presentations, bladder cancer has been classified into two different types: a frequently recurring papillary tumor (Ta), and a more aggressive carcinoma in situ (CIS). Either type can progress into invasive tumors (T1–T4), and the chance that low-grade Ta tumors develop into invasive disease is much less likely than that for high-grade Ta tumors and CIS. Of note, nowadays, it is commonly accepted that noninvasive tumors have been grouped as ‘non-muscle invasive bladder cancer’ and that noninvasive (Ta and CIS) tumors are distinguished from invasive tumors, which invade the basement membrane [17,18,19].

Loss of heterozygosity (LOH) is typically identified by comparing the DNA isolated from tumors to normal DNA, such as that isolated from blood, the germline control. This LOH can be detected using a method known as microsatellite instability analysis (MSA) [1,2,3]. LOH involving the p16 (chromosome 9p21) and p53 (17p13) are associated with non-muscle-invasive transitional cell carcinoma (TCC) and are so far known to be the two most common regions of LOH in bladder cancer [20,21,22,23,24,25]. LOH at 9p has been shown to have prognostic value in non-muscle-invasive bladder cancer, while other LOH loci implicated in bladder cancer progression are characterized among different genetic loci, including 18q, 4p, 16, 20, and 21 [26,27,28,29,30,31,32,33,34]. LOH is typically identified by comparing the DNA isolated from tumors to normal DNA, such as that isolated from blood or buccal swabs, which are germline control. Short tandem repeat (STR) regions, also known as microsatellite regions within chromosomes, are unstable in cancerous cells and are deleted by errors made during DNA replication, causing an LOH in the tumor sample. Microsatellite analysis (MSA) targets such tandem repeats in genomic DNA and is primarily designed to evaluate LOH that occurs with tumor cell transformation [22,23,24]. An initial report from Johns Hopkins University used a panel of fifteen STR, or microsatellite regions. It demonstrated that this MSA can detect deletions in the DNA isolated from urine sediment of bladder cancer patients before the cystoscopy-based detection of tumors [33,34,35]. Several follow-up studies have shown that these microsatellite changes can be profiled in urine to detect bladder cancer cells [36,37,38,39,40,41,42,43]. Overall, microsatellite analysis (MSA) combines 15 to 20 markers from a region with a high percentage of LOH. Various combinations of MSI markers so far resulted in overly sensitive defection for low- and high-grade lesions with sensitivities of 67%, 86%, and 93% for recurrent G1, G2, and G3 lesions, respectively, while such MSA still carries an average specificity of 88% [36,37,38,39,40,41,42,43,44]. Moreover, microsatellite analysis (MSA) could predict bladder tumor recurrence before cystoscopically detecting recurrent disease in all studies with extended follow-up [42,43,44]. Although MSA has an extremely high potential, several important issues and concerns have been raised and need to be resolved before their use in day-by-day clinical practice. These, including the establishment of assay automation, completion of multi-center studies, and establishment of guidelines excluding patients with persistent leukocyturia, need to be properly addressed if this analysis is to become clinically important. While LOH is typically identified by comparing the DNA isolated from tumors to germline DNA, microsatellite instability and LOH are two different genomic instability types; one concerns microsatellites, and the other is chromosomal. Moreover, their etiology is different. For example, tumors with microsatellite instability in colorectal tumors carry a completely different pathogenesis and outcome compared to chromosomal instability tumors [8]. Overall, LOH can be detected by gene scan, which may differ from microsatellite instability analysis [6].

Biomarkers have emerged as one of the most crucial elements in the early detection of cancer. The process of validating biomarkers to move them from the laboratory to the clinic poses technical and interpretive challenges. Hence, rigorous quality control is vital to address these challenges during assay validation. Early Detection Research Network (EDRN), an initiative of the National Cancer Institute (NCI), was recently established to discover, validate, and translate biomarkers for the early detection of cancer. As one of its initial vital projects, a prospective study to determine the efficacy of a novel set of biomarkers for the early detection of bladder cancer using MSA was funded by the EDRN and performed through a collaboration between academia and industry partners. MSA has been considered to be a promising technique recently developed into a clinical molecular assay for the surveillance of bladder cancer, and the prospective study was designed to determine the efficacy of the panel of MSA markers for detecting bladder cancer and bladder cancer recurrence using the developed clinical MSA assay [45,46]. The goals of this study could be summarized into three different areas: (1) to determine the sensitivity and specificity of the MSA assay on urine sediment as compared to cystoscopy and urine cytology; (2) to determine the temporal performance characteristics of the assay in urine sediment; (3) to determine the value of the MSA assay for early detection and recurrence of bladder cancer. As a part of the study, a quality control CLIA/College of American Pathology (CAP) accredited laboratory (QA Lab), University of Maryland Baltimore Biomarker Reference Laboratory (UMB-BRL), performed quality control analysis on 10% of the samples performed by the testing laboratory. Quality assurance measures included a concordance study between the testing laboratory (AIBioTech), CLIA/CAP accredited, and the QA lab to ensure that the assay was performed and the results were analyzed consistently. Therefore, a series of qualification studies were performed following the technical transfer and training of the MSA assay to the UMB-BRL and before the testing lab’s analysis of the clinical samples. This report details the steps taken to ensure assay qualification and illustrates the technical challenges facing biomarker validation.

## 2. Results

Among 9 carefully selected, high-quality publications outlined in Table 1, we have chosen 16 different combinations of MSA primer sets, as in Table 2. These primer sets have repeatedly shown a high LOH among bladder cancer samples while demonstrating a lack of LOH in normal samples. From over 350 patients, these 16 different combinations of MSA primer sets have provided over 90% sensitivity and 100% specificity. Our study aimed to develop dependable, accurate, and less costly tests. Initially, single plex with D13S304 ((AG)n 6-FAM-GGCTGCATGAGCCCTAAGTA TGGGTGACACAGTGAGACTCTA) was not included, as this marker had resulted in only inconsistent PCR amplification throughout repeated testing. Therefore, we dropped its use as a part of our final triplex PCR assay. MP2 marker MBP generated two distinct amplifications included in the results as marker MBP and MBPA, making 16 amplified markers from 15 primer sets in the multiplex (Table 2A,C). The cut-off value from this final triplet assay was determined and then repeated multiple times on various samples, including the original 25 cancer samples and 15 normal samples, and reproducibility was consistent (Table 2B).

### Development of Triplet Multiplex PCR

To keep testing parameters as consistent as possible between the two laboratories, the testing lab provided the QA lab with extracted DNA and all reagents, including probes and primers for MSA detection, the standard operating procedure, PCR reagents, and positive control DNA. The probes and primers comprising the MSA (Table 2C) were combined into three multiplexes and one single plex reaction. Following amplification of matched blood and urine sediment DNA in the multiplex PCR reactions, the PCR fragments were resolved by capillary electrophoresis. The samples were blinded at the testing lab before delivery of the reagents and split samples to the QA Lab. The ratio of the peak heights of heterozygous urine alleles was compared to the ratio of peak height in the urine alleles, and the detection of loss of heterozygosity was identified by a comparison with previously determined cutoffs for normal vs LOH. All data generated by both laboratories were delivered to the DMCC for analysis of concordance of both individual locus calls and overall determination of whether a sample was positive or negative for LOH. For the first round of qualification (concordance study), even matched pairs of samples were tested; the concordance in the calls between the two laboratories was 78%. This result indicated technical and interpretive issues in the assay performance that needed to be resolved to improve concordance. Both laboratories performed a stepwise review of assay instrumentation, assay setup, reagent handling and storage, and SOP interpretation. While minor errors in reporting were due to clerical or SOP interpretive issues, the major finding was that the difference between the pre-mixed primer sets used by each laboratory (e.g., date of preparation, lot number, length of storage) differed between the two laboratories. Therefore, for the second round of qualification (concordance study), both laboratories prepared the primer multiplex mixes before testing. These primer mixes were tested on 20 matched pairs of blood and urine DNA samples. The overall concordance in calls (normal vs positive for LOH) for this round was 73% (233 of 319 loci calls identical) (Table 3). In total, 17 of 20 samples were called identically by the lab (85%). These results were deemed unacceptable as a qualification of the assay, and a technical review was performed by both laboratories. Additional interpretive issues were quickly identified; slight differences in the amplification efficiency or electrophoresis between the two laboratories led to differences in the interpretation of the analyst as to the presence or absence of LOH. To attempt to implement more consistency in interpretation, quantitative acceptance criteria were established for allele characteristics, such as peak height and ratio of one allele to its heterozygote, in order to reproducibly determine what constitutes a valid allele vs an artifact, stutter allele, or poorly amplified product. Additional modifications included a more robust positive control cell line that was more representative of the STRs being tested and revision of the cutoff ratios between blood and urine pairs to determine positive vs negative results.

Using the revised SOP and interpretive guidelines, a third round of qualification was performed using 22 blinded-matched pairs of blood and urine sediment DNA (Table 4). This third round of concordance resulted in calls between the two laboratories 87% (305/352 calls identical). However, only 15 of the 22 sample pairs were called identically between the two laboratories. Four of the seven non-concordant samples had a single positive allele, calling those samples positive, whereas the second laboratory called those samples normal. Moving forward, borderline results (i.e., one positive allele) were verified by single plex PCR analysis of the positive locus. The sample was considered positive if the locus was positive for LOH in the single plex reaction. One sample pair, sample 37, had a single informative allele according to one laboratory, whereas the entire assay failed in the second lab. Criteria were, therefore, established for the number of informative alleles for each sample that would constitute a valid assay. These minor modifications in the assay increased the concordance between the laboratories to 90%, with 20 of the 22 samples reported identically as positive or negative for LOH. Commercialization of the assay called for the introduction of a different PCR master mix (Roche FastStart Taqman Master Probe) and the use of HPLC-purified primers in the assay. This major modification to the SOP required an additional round of tests to qualify the new procedure.

Twenty blind-matched blood and urine genomic DNA pairs were tested for the fourth qualification round (Table 5). The concordance between the two laboratories decreased to 73% due to the introduction of the new master mix, which was amplified with such high efficiency that the peaks were off-scale using the electrophoresis conditions outlined in the SOP. This significant effect on the assay required an additional un-blinded study between the two laboratories to refine electrophoresis conditions and determine conditions under which re-injection of the samples would occur. It was also determined at this time that the D13S802 marker did not add value to the assay; of the 72 specimens tested thus far in the qualification, only 2 generated a positive LOH result at this locus. Therefore, this marker was dropped from the panel for future testing.

Following this re-tooling of the assay, a fifth round of qualification was performed using the new electrophoresis conditions (Table 6). Twenty matched pairs of blood and urine DNA were tested; the concordance between the two laboratories was 92%; the concordance for loci that were evaluable for both laboratories was 96%. Further, the overall accuracy evaluation of the samples (positive or negative for LOH) was 100% identical (20/20 sample).

## 3. Discussion

While many aspects of clinical presentations, including the expected clinical course of disease progression, make bladder cancer a potential screening target, the common consensus at this point, based on several careful analyses, is screening high-risk populations [13,14,15,16,17], not general populations. Indeed, there are two key considerations making bladder cancer screening important in the upcoming decades. Foremost, the persistently high prevalence of smoking in the last several decades is expected to contribute as a key hazard to long-term urothelial carcinogenic effects for the upcoming several generations. Moreover, as bladder cancer usually cannot metastasize before it becomes locally invasive [19,20,21], this poses a valuable opportunity for early detection of bladder cancer in the window of time between tumor origination and invasion. On average, 70–80% of bladder cancers are noninvasive, and groups with noninvasive or organ-confined invasive tumors (T1/2, N−) have higher survival rates than those with more advanced stage tumors, including extravesical tumors (T3/4, N−), lymph node metastases (any T, N+), or metastases to distant organs (any T/N, M+) [12,13,14,52,53,54,55,56,57,58,59]. It is unfortunate that while there have been numerous efforts for constant improvements in the management of these more advanced diseases using chemotherapy or chemo-radiation therapy, the overall management outcome of bladder cancer has not been significantly improved over the past three decades [1,2,3]. In summary, patients with organ-confined disease have a 92.5% chance of survival for five years, while current five-year survival rates for patients with extravesical disease/nodal metastasis and distant metastasis are much less, 44.7% and 6.1%, respectively. These findings suggested that once bladder cancer progresses beyond a certain stage, like organ-confined disease, successful management becomes significantly more difficult. Therefore, early detection of bladder cancer has been proposed to be the most effective way of improving its management outcome. Indeed, the management of noninvasive cancers does not usually depend on aggressive treatment measures like cystectomy, systemic chemotherapy, or chemoradiation therapy and, therefore, is associated with fewer treatment-related morbidities and is more effective than that of invasive tumors [16,52,53,54].

MSA for cancer detection has several significant technical challenges, specifically in allele calling and interpretation. The results mirror those obtained by Butler et al. during validation studies for STR analysis for their study of human identity. Most importantly, variability between the labs due to instrumentation, personnel skills, and operation protocol created differences in assay performance. Moreover, both stochastic effects and variations in peak height effects between the DNA derived from urine sediment samples and DNA derived from blood samples resulted in more variation in the final reports, adding to interpretation differences. These differences were especially problematic in testing samples that produced results slightly above or below the cut-off ratios established for LOH. Hence, establishing the parameters for determining LOH from potential tumor cells isolated from urine sediment becomes an essential challenge for properly qualifying the STR assay. LOH is ordinary in human solid tumors and allows the expressivity of recessive loss-of-function mutations in tumor suppressor genes [22,23]; therefore, detecting recurrent LOH in a genomic region can provide critical clues and future guidance for the localization of tumor suppressor genes. Multiple factors need to be considered for a proper interpretation of LOH. For example, most clinical samples collected from urine will contain a mixture of tumor and normal cells, producing a potential mixture at each locus being analyzed, and this, then, can obscure losses of genetic material resulting in tumor cells, while LOH can only be determined by comparative analysis of a control profile obtained from a blood sample. Recently, our group successfully reproduced a dependable 15 marker based on three multiplex PCR assays. Overall, this report illustrates the challenge of qualifying a technically complex biomarker assay that requires interpretation by an analyst.

So far, many independent groups have previously confirmed that MSA has superior sensitivity (75–96%) compared to cytology (13–50%) in various clinical settings [23,24,25,26,27,28,29,30,31,32,33,34,38,39,40,41,42,43,44,45,46,52,53,54,55,60]. In summary, our careful analysis based on 377 cancer samples from 1997 to 2001 reports suggested that the sensitivity of MSA was 90 percent and specificity was 100 percent [31,32,33,34,35,36,37,38,39,40,41]. Importantly, unlike conventional cytology [55,56,57,58,59], it appears that microsatellite analysis (MSA) can detect low-grade and low-stage disease as accurately as high-grade and high-stage disease [36,37,38,39,40,41,42,43,44,47,48,49,51,61,62,63,64,65,66,67,68]. Frigerio et al. [37] made one of the first exciting reports showing that the combined use of cytology and LOH analysis had resulted in higher sensitivity than either test alone for identifying primary tumors and could detect almost all recurrent diseases in voided urine. Moreover, van Rhijn et al. [43] have demonstrated that out of 47 Caucasian patients with confirmed non-muscle-invasive bladder TCC (37 pTa, 10 pT1) at initial diagnosis, MSA can potentially provide an efficient early screening tool. In summary, this report provided three crucial clinical observations. First, MSA correctly identified 94% (44/47) of primary tumors. Second MSA correctly identified 92% (12/13) of tumor recurrences. Third, MSA predicted the chance of future recurrences, as 75% (9/12) of tumor recurrences were molecularly detected 1–9 months before cystoscopy evidence of recurrent disease. Likewise, several important studies demonstrated that urine microsatellite analysis reliably detects non-muscle-invasive bladder tumors and can be used as a reliable test for detecting and predicting tumor recurrences before cystoscopy evidence of recurrent disease and that such a detection rate can be improved when offered in combination with urine cytology. In this sense, it is exciting to recognize that meta-analysis was carried out by van Rhijn et al. who had written one of the first extensive literature reviews for the use of 18 markers and concluded that MSA, ImmunoCyt, NMP22, CYFRA21-1, LewisX, and FISH are the most promising markers for surveillance [67,69,70,71,72,73,74,75,76,77] of early detection for bladder cancer. Nevertheless, two points must be carefully considered for this biomarker-based bladder cancer detection. First, insufficient clinical evidence warrants the substitution of the cystoscopy follow-up scheme for any currently available urine marker tests. Second, data so far are not consistent with the sole use of molecular tests in patients with a high risk of developing bladder cancer. In summary, many studies have shown that various molecular tests carry their best value in improving the diagnostic accuracy of high-risk groups of the population in the initial diagnosis of bladder cancer and predicting recurrence, only when used in conjunction with cytology and cystoscopy. It was further suggested that molecular testing can reduce the need for these procedures with cytology and cystoscopy.

Our report illustrates the challenge of qualifying a technically complex biomarker assay that requires interpretation by an analyst. While both laboratories performed the assay with identical reagents, samples, and equipment, occasionally, different results were generated due to slight variations in how the analyst interpreted the electropherograms. Without strict guidelines for peak heights, peak ratios, and several interpretable alleles, one laboratory’s interpretation of the data differed from the other. The institution of strict quantitative values that assisted the analysts in their interpretations increased the number of concordant calls between the laboratory’s operator efficiency, which may improve bladder cancer management.

Of note, our reports do lack several important ancillary pieces of information. We were not provided with clinicopathological information on the patient for any of the samples we used in this report. This paper can be improved with more detailed data, including age, sex, smoking history, clinical stage of the disease, and whether the tumors are muscle-invasive or non-invasive. Also, the MSI status of the cancer samples could not be tested either by MSA or by examining the expression of MMR proteins (hMLH1, hPMS2, hMSH2, and hMSH6) to validate our data further, as we did not have access to proper samples. Likewise, we did not have access to clinical information regarding the disease progression history of the patients who tested positive for cancer by MSA. We feel that with access to proper samples and their clinical information in addition to an increased number of cancer samples, we could have enhanced the quality of our report. When this study was initiated, at the stage of clinical trial protocol formation, it was felt that a small number of samples with a limited amount of clinical information were deemed enough for initial assay development with quality control studies, which were later used to analyze a more significant number of samples from the EDRN study.

Here, we want to point out several points to maximize the sensitivity and specificity of MSA, which were also recognized in other studies as areas of future improvement. First, establishing a robust genetic marker profile and determining individual threshold values for LOH/allelic imbalance for each of the analyzed microsatellites enabled us to maximize the sensitivity of tumor DNA detection without compromising its specificity. We have accomplished this outcome using rigorous statistical analysis based on reported high-quality publications. Second, we have used a careful set-up of dependable methods, which was necessary to avoid erroneous LOH judgments due to PCR artifacts repeatedly described by others. When we designed two-tube-based PCR covering 16 markers, it seemed to be initially successful. Then, our internal repeated testing caused problems in some marker amplification, and, therefore, we decided not to pursue this assay. Third, we performed MSA on a genetic analyzer, which is commonly used nowadays and was based on strict guidelines. As a standard method of MSA practice, this was crucial in our quality control assay, particularly for the interpretation of data and primer mixture preparations. Therefore, we propose that sample processing and reaction mixture preparation should be performed under strict guidelines. At the same time, the genetic analyzer-based approach can be primarily automatized. Results need to be provided as a numerical data readout to ensure independence of inter-observer variability associated with the complex interpretation of morphological features. They also need standard guidelines like the final interpretation of test findings, which can be reported as “interpretable”, “non-measurable”, or “non-interpretable.” These cautious interpretations based on rigorous guidelines will make it possible for LOH ratios to be determined on a platform with high reproducibility which can thus provide reliable results. We also hope to point out that standard genetic analyzers like the ABI 3100 machine give MSA a significant advantage, as MSA is little affected by changes in PCR conditions or amounts of genomic DNA applied when cell conservation is suboptimal for cytological evaluation and/or FISH. Notably, for our quality analysis report, earlier involvement of the QC laboratory in interpreting and analyzing the data would have likely improved the results of the initial qualification studies. The study may have benefitted from performing early unblinded parallel studies between the QC and testing laboratories to uncover interpretive and technical issues sooner. Further development of microsatellite testing is needed to improve specificity and the speed of testing and may improve bladder cancer management. In conclusion, we have presented the data for the development of our MSA, a quality control assay. The CLIA/College of American Pathology (CAP) accredited laboratory, (QA Lab), University of Maryland Baltimore Biomarker Reference Laboratory (UMB-BRL), performed quality assurance analysis. Quality assurance measures included a concordance study between the testing laboratory (AIBioTech), CLIA/CAP accredited, and the QA lab to ensure that the assay was performed and the results were analyzed consistently. This report details the steps to ensure assay qualification and illustrates the technical challenges facing biomarker validation. We have discussed various prior data regarding MSA-based bladder cancer detection along the line. We hope our assay can potentially improve the management of bladder cancer [78,79,80].

## 4. Materials and Methods

### 4.1. Matched Blood and Urine Genomic DNA Samples

Blood and urine specimens for the qualification study were obtained from Dr David Sidransky’lab at Johns Hopkins University. These included 15 biopsy-proved non-muscle-invasive bladder cancer and 10 normal controls, and participants in the assay development for the MSA study we report here were all deemed eligible for the study. These samples were previously used for the determination of a proper cutoff value for a series of 10 MSA markers.

### 4.2. DNA Extraction and Quantification

Matched blood and urine samples remained de-identified during the study. Genomic DNA from the buffy coat of blood collected in ACD collection tubes was purified by the testing lab using the QIAGEN QIAMP Blood Mini kit according to manufacturer’s instructions. Genomic DNA from urine sediment was purified using the QIAGEN QIAMP Viral RNA Mini kit. The DNA was quantified against a standard curve of human DNA using TaqMan β-actin Detection Reagents from Applied Biosystems. Amounts of 30 mL of urine samples and 5 mL of whole blood were used for this study.

### 4.3. STR Targets and PCR Primers

The panel of STR targets for the MSA assay was originally developed at Johns Hopkins University as a radioactive PCR assay [23]. The testing lab converted the radioactive PCR assay to a high-throughput capillary electrophoresis assay of fluorescent PCR products (Table 2). The resulting assay was comprised of two or three multiplex reactions. The 5′ fluorescent primers were obtained from Applied Biosystems. The 3′ primers were obtained from Integrated DNA technologies. Following round three of the qualification studies, the single plex reaction was dropped due to poor performance in the assay. The primers were initially mixed at an equimolar concentration (2 pm each primer); following refinement of the assay, the primer concentrations were adjusted. The five sets of multiplex PCR were previously determined from an assay developed for EDRN study [36,37].

### 4.4. Multiplex PCR: Triplet (Three Tube) Multiplex PCR

For the first three rounds of qualification, the STR regions were amplified using AmpliTaq Gold DNA Polymerase Applied Biosystems). The PCR conditions were 4 mM MgCl_2_/0.2 mM dCTP/0.2 mM dGTP/0.4 mM dUTP, 0.5 mM dATP/2 units Amplitaq Gold. MP1, MP2 were amplified using 4 ng of total genomic DNA, respectively, for both blood and urine DNA, and MP3 and the D13S804 Singleplex reaction were amplified using 6 ng of blood or urine DNA in a 25 µL final volume (Table 2C). The primer compositions for each locus are listed in Table 2A and 2B. PCR cycling conditions were 95 °C for 11 min/32 cycles of 94 °C 1 min, 55 °C for 1 min, 72 °C for 1 min/60 °C for 45 min/4 °C hold. For rounds 4 and 5 of the qualification, the PCR amplification conditions were modified to utilize 1X FastStart Taqman Probe Master (Roche) with various primer concentrations. The DNA concentrations and the PCR cycling conditions remained the same.

### 4.5. Capillary Electrophoresis

Capillary electrophoresis was performed on 1 µL of PCR product using the 3100 Genetic analyzer from Applied Biosystems (Foster City, CA, USA). Matched blood and urine samples were run in the same injection to prevent differences in run conditions between matched specimens. Following electrophoresis, the data were analyzed using Gene mapper v3.1 (Applied Biosystems), and the peak sizes and peak heights were exported as a tab delimited text files.

### 4.6. Calculation of Loss of Heterozygosity

The data exported from the 3100 analyzer were imported in Microsoft Excel for analysis. In order for the data to be acceptable, the positive control DNA (HL-60 genomic DNA (ATCC)) had to pass strict peak height and size criteria before the sample data could be analyzed. If these criteria were acceptable, the ratio of the blood to urine heterozygous alleles used the following formula: ratio = (urine 1 allele 1 peak height/urine 1 allele 2 peak height)/(blood 1 allele 1 peak height/blood 1 allele 2 peak height). This ratio was then compared to cutoff values previously determined by the calculation of ratios of 50 matched normal blood and urine specimens.

### 4.7. Data Analysis

Data analysis was performed by GeneMapperID; “Positive” results indicated a variant or missing allele at the loci indicated. Target loci are known cancer markers. LOH at the loci indicated when one of the two alleles was missing in the urine sample at a heterozygous locus in the buccal sample. LOH can only be detected at heterozygous loci; hence, homozygous loci are uninformative. The overall MSA genotype was calculated as negative; LOH—low (1–2 markers), LOH—medium (3–4 markers), and LOH—high (>4 markers). Tumors typically acquire more LOH markers as they grow and advance in stage. Unless stated otherwise in the validation protocol, the following acceptance criteria were applied to determine the suitability of each assay:

### 4.8. ABI 3100 DNA Analyzer Acceptance Criteria

To be deemed negative (or no LOH detected), a heterozygous locus must have RFU peak heights between 200 RFUs and 100,000 RFUs with a peak height ratio between the cut off values determined for each STR marker (see Table 2). To be deemed LOH/positive, the samples must have at least one heterozygous loci with RFU peak heights that fall between 200 RFUs and 100,000 RFUs with a peak height ratio above or below the cut off values determined for each STR marker (see Table 2). To be deemed non-informative, the sample has to be homozygous for the locus. Homozygous locus will have only one peak with a sample RFU peak height that falls between 200 RFUs and 100,000 RFUs. If the sample does not meet the minimum or maximum RFU for peak height or has no signal at all, it will be deemed non-evaluable. A minimum of eight (8) loci must be analyzed and meet the negative, LOH, or non-informative call in order to pass the sample acceptance criteria. Failed samples will be repeated one time. If a sample fails on the second analysis, then it is deemed to have failed the sample acceptance criteria and is reported as Quantity Not Sufficient (QNS).

## Figures and Tables

**Table 1 ijms-25-00209-t001:** List of prior publications for selection of 16 markers.

Scheme.	No. of Cancers Detected by MSA	Sensitivity (%)	Healthy Controls with Neg MSA Result	Specificity (%)
Mao et al. [34]	19/20	95	5 out of 5	100
Steiner et al. [42]	10/11	91	10 out of 10	100
Linn et al. [47]	13/15	87	N/A	N/A
Schneider et al. [48]	87/103	84	N/A	N/A
Sourvinos et al. [49]	26/28	93	10 out of 10	100
Zhang et al. [50]	73/81	90	19/19	100
Seripa et al. [36]	33/34	97	11 out of 11	100
Zhang et al. [51]	22/23	96	17/17	100
Amira et al. [44]	44/47	94	N/A	N/A
Overall	327/362	90%	72/72	100%

MSA, microsatellite instability analysis; N/A, not available.

**Table 2 ijms-25-00209-t002:** (A) upper panel: list of microsatellite instability (MSI) forward and reverse primers with their DNA sequence, their repeats, size range, and color channels. FGA denotes the gene making a protein fibrinogen A, MBP for myelin basic protein, TH01 for tyrosine hydroxylase 1, and IFNA1 for in-terferon alpha 1. (B) lower panel: list of microsatellite instability (MSI) markers for their size cut off range. (C): mixtures for combinations of primers for triplex PCR reaction.

**2A**					
**Locus**	**Repeat Type**	**Size Range**	**Channel**	**K562-Allele Sizes**	
		**Color**			
D4S243	(ATAG)n	165–192 bp	Blue	169 bp	
D4S243					
FGA	(TTTC)n	299–361 bp	Green	328 bp	
D9S747	(GATA)n	179–201 bp	Green	185 bp	
Dl 7S654	(CA)n	194–218 bp	Yellow	216 bp	
D9S162	(CA)n	117–148 bp	Yellow	143 bp	
D17S695	(AAAG)n	170–220 bp	Red	185, 200 bp	
MBP & A	(ATGG)n	200–242/119–151	Blue	207, 215, 119 bp	
D21S1245	(AAAG)n	209–293 bp	Green	236, 255 bp	
D16S310	(ATAG)n	127–170 bp	Green	155, 160 bp	
D20S48	(GT)n	251–269 bp	Yellow	261 bp	
THOl	(TCAT)n	174–209 bp	Yellow	198 bp	
D9Sl 71	(CA)n	109–129 bp	Yellow	126 bp	
D16S476	(AAAG)n	176–230 bp	Red	187 209 bp	
IFN-A	(GT)n	132–152 bp	Red	no product	
**2B**					
**Marker**	**Lower Limit**		**Upper Limit**		
D4S243	0.68		1.33		
FGA	0.6		1.43		
D9S747	0.63		1.42		
Dl 7S654	0.71		1.36		
DI 7S695	0.49		1.61		
MBP	0.63		1.45		
MBPA	0.71		1.37		
D16S310	0.6		1.34		
D9S162	0.51		1.53		
THOl	0.46		1.53		
IFN-A	0.68		1.43		
D21Sl245	0.58		1.42		
D20S48	0.62		1.46		
D9Sl 71	0.72		1.36		
D16S476	0.54		1.65		
**2C**					
**Multiplex**	**Target**	**Chromosome**	**STR**	**Forward Primer Sequence (5′-3′)**	**Reverse Primer Sequence (5′-3′)**
MP1	D4S243	4	(ATAG)n	6-FAM-TCAGTCTCTCTTTCTCCTTGCA	TAGGAGCCTGTGGTCCTGTT
	FGA	4	(TTTC)n	VIC-GACATCTTAACTGGCATTCATGG	CTTCTCAGATCCTCTGACACTCG
	D9S747	9	(GATA)n	VIC-GCCATTATTGACTCTGGAAAAGAC	CAGGCTCTCAAAATATGAACAAAAT
	D17S654	17	(CA)n	NED-ACCTAGGCCATGTTCACAGC	GAGCAGAATGAGAGGCCAAG
	D17S695	16	(AAAG)n	PET-CTGGGCAACAAGAGCAAAAT	TTTGTTGTTGTTCATTGACTTCAGTC
MP2	D9S162	9	(CA)n	NED-GCAACCATTTATGTGGTTAGGG	TCCCACAACAAATCTCCTCAC
	MBP	18	(ATGG)n	6-FAM-GGACCTCGTGAATTACAATCACT	ATCCATTTACCTACCTGTTCATCC
	D16S310	16	(ATAG)n	VIC-GGGCAACAAGGAGAGACTCT	AAAAAAGGACCTGCCTTTATCC
	THO1	11	(TCAT)n	NED-AGGCTCTAGCAGCAGCTCAT	TGTACACAGGGCTTCCGAGT
	IFN-A	9	(GT)n	PET-TGCGCGTTAAGTTAATTGGTT	GTAAGGTGGAAACCCCCACT
MP3	D21S1245	21	(AAAG)n	VIC-CCAGAAAATGACACATGAAGGA	TTGTTGAGGATTTTTGCATCA
	D20S48	20	(GT)n	NED-ATGGTCTCCAGTCCCATCTG	TTGACCTGGATGAGCATGTG
	D9S171	9	(CA)n	NED-TCTGTCTGCTGCCTCCTACA	GATCCTATTTTTCTTGGGGCTA
	D16S476	16	(AAAG)n	6-FAM-GGCAACAAGAGCAAAACTCC	GGTGCTCTCTGCCCTATCTG

PCR Primer pairs for MSA assay. Three multiplexes (MP1, MP2, MP3) were designed to amplify fifteen targets. The 5′ primers are differentially labeled to enable detection by different fluorophores.

**Table 3 ijms-25-00209-t003:** Concordance study results from the second round of analysis.

	Sample Number																				
	Marker	1	2	3	4	5	6	7	8	9	10	11	12	13	14	15	16	17	18	19	20
MP1	D4S243	11	11	22	11	22	13	11	22	11	22	33	22	11	11	21	33	11	22	11	11
	FGA	22	11	22	11	11	22	11	13	11	11	22	22	11	11	11	13	11	11	11	11
	D9S747	11	11	22	11	11	11	22	22	22	11	33	22	22	11	22	11	11	22	11	22
	D17S654	11	11	22	22	11	11	21	21	11	21	33	22	21	21	11	33	13	11	11	31
	D17S695	11	11	22	22	22	12	11	12	12	22	33	22	11	12	12	32	22	11	12	11
MP2	MBP	11	11	22	11	22	11	11	33	11	11	33	22	11	11	11	33	11	11	11	11
	MBPa	11	11	22	11	22	11	11	33	11	11	22	22	13	11	21	33	11	11	22	11
	D16S310	22	22	22	22	22	11	21	11	11	22	22	22	22	22	22	22	22	11	22	22
	TH01	11	21	22	12	11	12	11	21	11	12	23	22	13	11	21	11	12	21	11	11
	D9S162	11	12	22	21	11	21	11	12	11	21	32	22	11	11	32	11	21	12	11	11
	IFN-A	21	22	22	21	22	11	22	22	22	21	22	22	22	22	13	13	11	41	22	11
MP3	D21S1245	12	11	22	12	11	12	11	33	22	12	22	22	33	22	13	33	22	11	11	11
	D20S48	11	21	22	11	11	11	21	21	11	21	23	22	21	11	11	23	11	11	21	21
	D9S171	11	21	22	22	11	11	21	11	11	12	11	22	22	21	13	23	12	12	22	11
	D16S476	12	11	22	22	11	12	12	12	22	22	32	22	12	12	12	21	21	12	12	12
SP	D13SB02	21	22	22	12	22	21	21	22	22	22	22	22	22	22	22	32	21	11	22	22
	AIB	1	1	2	1	1	1	1	3	1	1	3	2	3	1	3	3	1	1	1	3
	UMMC	1	1	2	1	1	3	1	3	1	1	3	2	3	1	3	3	3	1	1	1
	Agreement	Y	Y	Y	Y	Y	N	Y	Y	Y	Y	Y	Y	Y	Y	Y	Y	N	Y	Y	N

Concodance following the second round of qualification. Samples evaluated by The testing lab, AIB (first number) and UMMC (second number) are shown. 11 means 1(AIB) and 1 (UMMC). 1 = Negative; 2 = Non-informative; 3 = Positive for LOH; 4 = Non-evaluable. Loci that are not concordant are indicated in red color.

**Table 4 ijms-25-00209-t004:** Concordance study results from the third round of analysis.

	Sample Number																				
	Marker	1	2	3	4	5	6	7	8	9	10	11	12	13	14	15	16	17	18	19	20
MP1	D4S243	11	33	11	11	11	22	11	11	11	11	11	22	11	22	31	11	11	11	11	11
	FGA	11	33	11	11	13	11	11	11	11	11	11	11	11	11	22	11	13	11	11	22
	D9S747	11	33	11	11	22	31	22	11	31	22	22	22	33	11	11	11	11	11	11	11
	D17S654	21	33	11	31	11	31	11	11	22	11	11	11	22	22	31	22	33	11	11	11
	D17S695	21	33	11	22	11	31	11	11	11	11	11	11	22	22	21	11	33	22	22	11
MP2	MBP	22	33	11	11	11	31	11	11	31	11	11	11	11	11	11	22	33	22	11	11
	MBPa	11	33	11	11	22	22	22	11	31	11	11	22	11	22	11	22	33	22	11	11
	D16S310	11	12	11	11	11	11	22	11	31	13	11	11	11	11	11	11	11	11	22	11
	TH01	11	22	11	11	11	22	11	11	22	11	11	11	22	11	11	11	11	11	12	22
	D9S162	11	33	11	11	11	11	11	11	11	22	11	11	22	22	11	11	11	22	11	22
	IFN-A	11	22	22	13	22	22	22	22	31	11	11	11	33	11	11	22	11	22	22	22
MP3	D21S1245	22	22	21	11	21	22	22	11	21	21	11	22	21	11	21	21	11	11	11	11
	D20S48	11	11	11	22	11	11	11	22	11	22	11	11	11	11	11	11	11	11	11	11
	D9S171	11	33	11	22	13	11	11	22	32	13	11	11	33	22	22	11	22	22	22	11
	D16S476	21	22	11	22	11	11	22	11	11	11	11	11	11	22	11	22	11	22	22	11
SP	D13SB02	22	22	22	31	22	22	22	22	22	22	22	22	22	22	22	11	22	22	22	22
	AIB	1	3	1	3	1	3	1	1	3	1	1	1	3	1	3	1	3	1	1	1
	Maryland	1	3	1	3	3	1	1	1	1	3	1	1	3	1	1	1	3	1	1	1
	Agreement	Y	Y	Y	Y	N	N	Y	Y	N	N	Y	Y	Y	Y	N	Y	N	Y	Y	N

Concodance following the third round of qualification. Samples evaluated by The testing lab, AIB (first number) and UMMC (second number) are shown. 11 means 1 (AIB) and 1 (UMMC). 1 = Negative; 2 = Non-informative; 3 = Positive for LOH; 4 = Non-evaluable. Loci that are not concordant are indicated in red color.

**Table 5 ijms-25-00209-t005:** Concordance study result from the fourth round of analysis.

	Sample Number																				
	Marker	1	2	3	4	5	6	7	8	9	10	11	12	13	14	15	16	17	18	19	20
MP1	D4S243	11	33	11	11	11	22	11	11	11	11	11	22	11	22	31	11	11	11	11	11
	FGA	11	33	11	11	13	11	11	11	11	11	11	11	11	11	22	11	13	11	11	22
	D9S747	11	33	11	11	22	31	22	11	31	22	22	22	33	11	11	11	11	11	11	11
	D17S654	21	33	11	31	11	31	11	11	22	11	11	11	22	22	31	22	33	11	11	11
	D17S695	21	33	11	22	11	31	11	11	11	11	11	11	22	22	21	11	33	22	22	11
MP2	MBP	22	33	11	11	11	31	11	11	31	11	11	11	11	11	11	22	33	22	11	11
	MBPa	11	33	11	11	22	22	22	11	31	11	11	22	11	22	11	22	33	22	11	11
	D16S310	11	12	11	11	11	11	22	11	31	13	11	11	11	11	11	11	11	11	22	11
	TH01	11	22	11	11	11	22	11	11	22	11	11	11	22	11	11	11	11	11	12	22
	D9S162	11	33	11	11	11	11	11	11	11	22	11	11	22	22	11	11	11	22	11	22
	IFN-A	11	22	22	13	22	22	22	22	31	11	11	11	33	11	11	22	11	22	22	22
MP3	D21S1245	22	22	21	11	21	22	22	11	21	21	11	22	21	11	21	21	11	11	11	11
	D20S48	11	11	11	22	11	11	11	22	11	22	11	11	11	11	11	11	11	11	11	11
	D9S171	11	33	11	22	13	11	11	22	32	13	11	11	33	22	22	11	22	22	22	11
	D16S476	21	22	11	22	11	11	22	11	11	11	11	11	11	22	11	22	11	22	22	11
SP	D13SB02	22	22	22	31	22	22	22	22	22	22	22	22	22	22	22	11	22	22	22	22
	AIB	1	3	1	3	1	3	1	1	3	1	1	1	3	1	3	1	3	1	1	1
	Maryland	1	3	1	3	3	1	1	1	1	3	1	1	3	1	1	1	3	1	1	1
	Agreement	Y	Y	Y	Y	N	N	Y	Y	N	N	Y	Y	Y	Y	N	Y	N	Y	Y	N

Concodance following the fourth round of qualification. Samples evaluated by The testing lab, AIB (first number) and UMMC (second number) are shown. 11 means 1 (AIB) and 1 (UMMC). 1 = Negative; 2 = Non-informative; 3 = Positive for LOH; 4 = Non-evaluable. Loci that are not concordant are indicated in red color.

**Table 6 ijms-25-00209-t006:** Concordance study. Results from the fifth round of analysis.

	1BU		2BU		3BU		4BU		5BU		6BU		7BU		8BU		9BU		10BU		11BU		12BU		13BU		14BU		15BU		16BU		17BU		18BU		19BU		20BU	
Marker	A	M	A	M	A	M	A	M	A	M	A	M	A	M	A	M	A	M	A	M	A	M	A	M	A	M	A	M	A	M	A	M	A	M	A	M	A	M	A	M
D4S243	2	2	1	1	2	2	2	1	1	1	1	1	2	2	1	1	1	1	3	3	3	3	3	4	1	1	2	2	1	2	1	1	1	1	1	1	1	1	2	2
FGA	2	2	1	1	1	1	1	1	1	1	2	2	1	1	1	1	1	1	1	1	3	3	3	3	1	1	1	1	1	1	1	1	1	1	1	1	1	1	1	1
D9S747	2	2	2	2	1	1	1	3	1	1	4	1	1	1	1	1	1	1	3	3	2	2	3	3	1	1	1	1	1	1	1	1	1	1	1	1	3	3	3	2
D17S654	1	1	1	1	1	1	2	2	1	1	1	1	1	1	2	2	1	1	3	3	3	3	1	1	1	1	2	2	1	1	1	1	1	1	1	1	2	2	1	1
D17S695	1	1	1	1	1	1	1	1	1	1	2	1	1	1	1	1	1	1	3	3	2	2	1	1	1	1	2	2	1	1	2	2	4	2	1	1	2	2	1	1
MBP	1	1	1	1	2	2	3	3	3	3	1	1	1	1	1	1	1	1	3	4	2	2	3	3	1	1	1	1	1	1	1	1	2	2	1	1	1	1	1	1
MBPa	1	1	2	2	2	2	3	3	1	1	1	1	1	1	1	1	1	1	3	3	2	2	3	4	1	1	2	2	1	1	1	1	2	2	1	1	1	1	2	2
D16S310	1	1	1	1	1	1	1	1	2	2	1	1	4	1	1	1	1	1	2	2	1	1	3	1	1	1	1	1	1	1	1	1	1	1	1	4	1	1	1	1
TH01	1	1	1	1	1	1	2	2	1	1	1	1	1	1	1	1	1	1	2	2	3	3	3	3	1	1	1	1	1	1	1	1	1	1	1	1	2	2	1	1
D9S162	2	2	1	1	1	1	1	1	1	1	1	1	1	1	1	1	1	1	3	3	3	3	3	3	2	2	2	2	1	1	1	1	2	2	2	2	2	2	1	1
IFN-A	1	1	2	2	2	2	1	3	3	3	1	1	2	2	1	1	2	2	2	2	2	2	3	3	1	1	1	1	2	2	1	1	2	2	1	1	3	3	1	1
D21S1245	1	1	1	1	1	1	1	1	4	4	1	1	4	1	4	1	1	1	2	2	1	3	3	3	1	1	1	1	1	1	1	1	1	1	1	1	1	1	4	4
D20S48	2	2	1	1	1	1	1	1	3	4	1	1	1	1	1	1	1	1	1	1	2	2	3	3	2	2	1	1	1	1	2	2	1	1	1	1	1	1	1	1
D9S171	1	1	1	1	2	2	1	1	1	1	2	2	1	1	1	1	1	1	3	3	3	2	3	3	1	2	2	2	2	2	2	2	2	2	1	2	3	3	1	1
D16S476	1	1	1	1	2	2	1	1	1	1	1	1	1	1	1	1	1	1	2	2	1	1	3	3	1	1	1	2	2	2	2	2	2	2	1	1	1	1	1	1
Overall Evaluation	1	1	1	1	1	1	3	3	3	3	1	1	1	1	1	1	1	1	3	3	3	3	3	3	1	1	1	1	1	1	1	1	1	1	1	1	3	3	1	1

Concodance following the fifth round of qualification. Samples evaluated by The testing lab, AIB (first number) and UMMC (second number) are shown. 11 means 1 (AIB) and 1 (UMMC). 1 = Negative; 2 = Non-informative; 3 = Positive for LOH; 4 = Non-evaluable. Loci that are not concordant are indicated in red color.

## Data Availability

Data are contained within the article.

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
