# Peer review of "Qualification of the Microsatellite Instability Analysis (MSA) for Bladder Cancer Detection: The Technical Challenges of Concordance Analysis"

_ijms, 2023, doi:10.3390/ijms25010209_

Round 1

Reviewer 1 Report

Comments and Suggestions for Authors

1.     The abstract repeats lines 47-67.

2.     We no longer use superfical or papillary bladder cancer, as the nomenclature changed into non-muscle invasive and muscle invasive. Please refer to https://uroweb.org/guidelines/non-muscle-invasive-bladder-cancer/chapter/pathological-staging-and-classification-systems.

3.     The data presented should be supplemented with clinical information, most importantly pathological findings. Although the authors mentioned this limitation in the respective section, maybe the data could be obtained before publication.

4.     Please improve the resolution of table 2 and remove the double use ‘table 2’a/table 2b, rather ‘A’ and ‘B’ part and combine it with table 2c

5.     The language of the paper is informal in several parts, please revise it.

6.     Lines 230-235 you should either cite the source or add the data into the manuscript.

Comments on the Quality of English Language

as above.

Author Response

  1. The abstract repeats lines 47-67.

These paragraphs were modified to avoid repeated sentence, highlighted with green shadow

  1. We no longer use superfical or papillary bladder cancer, as the nomenclature changed into non-muscle invasive and muscle invasive. Please refer to https://uroweb.org/guidelines/non-muscle-invasive-bladder-cancer/chapter/pathological-staging-and-classification-systems.

These are changed, highlighted with green shadow.

  1. The data presented should be supplemented with clinical information, most importantly pathological findings. Although the authors mentioned this limitation in the respective section, maybe the data could be obtained before publication.

We have provided all of the information available to us based on the protocol and discussed this. We simply did not have that information and we apologize for this shortcoming.

  1. Please improve the resolution of table 2 and remove the double use ‘table 2’a/table 2b, rather ‘A’ and ‘B’ part and combine it with table 2c

      This has been improved based on advice, highlighted with green shadow.

  1. The language of the paper is informal in several parts, please revise it.

       The language gas been extensively revised as highlighted with yellow shade

  1. Lines 230-235 you should either cite the source or add the data into the manuscript.

 We have decided to delete this paragraph as it itself is another rather complex analysis with an independent manuscript.

Reviewer 2 Report

Comments and Suggestions for Authors

Line 94,95 „LOH is typically identified by comapring…such as isolated from peripheral blood“-repetition, no need to repeat it

Table 2A and 2B are figures of tables and they are blurry so please insert actual table instead of figure of table

Line 257 it says “minima” instead of minimal

Lines 283-287 not relevant, remove

From this study I am only convinced that using MSI testing in bladder cancer has its benefits, but I did not get the sense that this is good test to analyze MSI. What is a scientific value, what is new? 

Author Response

Line 94,95 „LOH is typically identified by comapring…such as isolated from peripheral blood“-repetition, no need to repeat it

                   This has been modified and highlighted with gray shadow

Table 2A and 2B are figures of tables and they are blurry so please insert actual table instead of figure of table

               This has been improved based on advice.

Line 257 it says “minima” instead of minimal

This has changed.

Lines 283-287 not relevant, remove

This has been removed.

From this study I am only convinced that using MSI testing in bladder cancer has its benefits, but I did not get the sense that this is good test to analyze MSI. What is a scientific value, what is new?

You are correct in that it is another test analyze MSI while it provide a proof of principle of applying MSI for clinical practice.

Reviewer 3 Report

Comments and Suggestions for Authors

The manuscript "Qualification of the Microsatellite Instability Analysis (MSA) for Bladder Cancer Detection: The Technical Challenges of Concordance Analysis" by Thomas Reynolds et al. holds potential in advancing the field of bladder cancer detection through Microsatellite Instability Analysis (MSA). However, there are areas that require attention to enhance the clarity and scientific rigor of the manuscript.

Comments: 

Line 43-44: Typos need to be corrected.

Line 78: Remove the extra ")".

Microsatellite analysis was referred to multiple times; however, its abbreviated format (MSA) was defined at Line 71 and Line 80.

Line 76-77 and Line 94-95: "LOH is typically identified by comparing the DNA isolated from tumors to germ line DNA, such as that isolated from blood." This statement was made twice.

Line 98: Add the missing reference for "colorectal cancer".

Line 99: Could you elaborate more on "gene scan" as compared to microsatellite instability analysis?

Line 131-134: There are a few small grammatical and stylistic improvements could be made to improve clarity. For example: "These primer sets have repeatedly shown a dependable high ...."

Line 318-330 and Line 262-274: The statement was repeated twice in the discussion section: "Microsatellite analysis for detection of cancer has a number of significant technical..." and "LOH is common in human solid tumors and allows..."

Line 405: Correct "while blood" to "whole blood".

The methodology sections: a more detailed description of the methods, including catalog number, should be provided to ensure reproducibility of the study.

Author Response

Line 43-44: Typos need to be corrected.

This has been corrected, overall language has been extensively revised highlighted with yellow shadow.

Line 78: Remove the extra ")".

This has been corrected.

Microsatellite analysis was referred to multiple times; however, its abbreviated format (MSA) was defined at Line 71 and Line 80.

This has been corrected, highlighted with purple shadow.

Line 76-77 and Line 94-95: "LOH is typically identified by comparing the DNA isolated from tumors to germ line DNA, such as that isolated from blood." This statement was made twice.

This has been corrected, highlighted with  shadow.

Line 98: Add the missing reference for "colorectal cancer".

This has been added, highlighted with shadow.

Line 99: Could you elaborate more on "gene scan" as compared to microsatellite instability analysis?

This has been added as a reference.

Line 131-134: There are a few small grammatical and stylistic improvements could be made to improve clarity. For example: "These primer sets have repeatedly shown a dependable high ...."

This has been corrected, highlighted with yellow shadow.

Line 318-330 and Line 262-274: The statement was repeated twice in the discussion section: "Microsatellite analysis for detection of cancer has a number of significant technical..." and "LOH is common in human solid tumors and allows..."

This has been corrected, highlighted with  shadow.

Line 405: Correct "while blood" to "whole blood".

This has been corrected, highlighted with shadow.

The methodology sections: a more detailed description of the methods, including catalog number, should be provided to ensure reproducibility of the study.

We do not usually keep catalogue number and we apologize for this

Round 2

Reviewer 1 Report

Comments and Suggestions for Authors

The authors responded to all my comments.

Author Response

No further comments